# Antigenotoxic Effect of Ascorbic Acid and Resveratrol in Erythrocytes of *Ambystoma mexicanum, Oreochromis niloticus* and Human Lymphocytes Exposed to Glyphosate

Carlos Alvarez-Moya [1],*[ID], Alexis Gerardo Sámano-León [1], Mónica Reynoso-Silva [1],*, Rafael Ramírez-Velasco [1], Mario Alberto Ruiz-López [2] and Alma Rosa Villalobos-Arámbula [3]

[1] Environmental Mutagenesis Laboratory, Cellular and Molecular Department, University of Guadalajara, Guadalajara 45200, Jalisco, Mexico; alexis.samano@alumnos.udg.mx (A.G.S.-L.); raravel96@gmail.com (R.R.-V.)

[2] Biotechnology Laboratory, Department of Botany and Zoology, University of Guadalajara, Guadalajara 45200, Jalisco, Mexico; mruiz@cucba.udg.mx

[3] Molecular and Functional Genetics and Ecosystems Laboratory of Biomarkers and Molecular Genetics, Cellular and Molecular Departament, University of Guadalajara, Guadalajara 45200, Jalisco, Mexico; avillal@cucba.udg.mx

[*] Correspondence: calvarez@cucba.udg.mx (C.A.-M.); monica.reynoso@cucba.udg.mx (M.R.-S.); Tel.: +52-377-77-1121 (C.A.-M.); +52-333-777-1121 (M.R.-S.)

**Abstract:** Glyphosate is a controversial herbicide. Its genotoxicity and presence in various ecosystems have been reported. The use of ascorbic acid and resveratrol could protect different organisms from glyphosate-induced genetic damage. In the present study, specific genetic damage induced by glyphosate was evaluated in erythrocytes of *Oreochromis niloticus*, *Ambystoma mexicanum* and human lymphocytes. Simultaneously, the antigenotoxic capacity of various concentrations of ascorbic acid and resveratrol was evaluated by means of pretreatment and simultaneous treatment protocols. The 0.03, 0.05 and 0.07 mM concentrations of glyphosate induced significant genotoxic activity ($p < 0.05$) in human lymphocytes and in erythrocytes of the species studied, and could cause genomic instability in these populations. The reduction in genetic damage observed in human lymphocytes exposed to high concentrations of glyphosate is only apparent: excessive genetic damage was associated with undetectable excessive tail migration length. A significant ($p < 0.05$) antigenotoxic effect of ascorbic acid and resveratrol was observed in all concentrations, organisms and protocols used. Both ascorbic acid and resveratrol play an important role in maintaining the integrity of DNA. Ascorbic acid in *Oreochromis niloticus*, *Ambystoma mexicanum* reduced glyphosate-induced genetic damage to a basal level. Therefore, our data indicate that these antioxidants could help preserve the integrity of the DNA of organisms exposed to glyphosate. The consumption of antioxidants is a useful tool against the genotoxicity of glyphosate.

**Keywords:** glyphosate; ascorbic acid; resveratrol; bioassays; genetic damage; DNA protection

## 1. Introduction

Pesticide exposure is a growing problem. Glyphosate or N (phosphonomethyl) glycine is one of the most widely used herbicides in the world and its application increased from 56,296 tons in 1994 to 825,804 in 2014 [1]. It is not normally applied in its pure chemical form and it is marketed as commercial glyphosate-based formulations that include adjuvants capable of improving its agricultural function [2]. The half-life of glyphosate is from 5 to 70 days, depending on the environment in which it is located. Its persistence in the environment and the increase in its use have made it necessary to evaluate the genotoxic effect in species of various ecosystems [3,4]. Recent reports warn that glyphosate has negative effects on both terrestrial and aquatic organisms [5–8], however, the genotoxicity of glyphosate

has been reported in invertebrates [9], fish and plants [10,11], amphibians [12–14], reptiles [15–17], birds [18], non-human mammals [19,20] and humans [2,21]. The genetic damage induced by glyphosate depends on the concentration and time of exposure [22,23], in addition, the use of adjuvants in commercial glyphosate-based formulations considerably increases the toxicity of the herbicide [22]. Glyphosate has been detected in blood of humans in a mean concentration of 0.0007–0.7 mM [10]. Exposure to glyphosate is associated with different types of cancer [24,25], adverse effects on the reproduction of rats and humans [26,27], promotion of carcinogenesis in rats [28], toxicity [6], liver and kidney damage [29]. Although the mechanism responsible for the genotoxicity of glyphosate is not well defined, it is speculated that glyphosate causes an increase in free radicals (ROS).

However, there exists a vigorous debate about genotoxicity and carcinogenicity of glyphosate by scientific and regulatory agencies [19]. Even more ignorance exists about its metabolism in the blood. Several test systems have been developed to identify alterations in the genome by genotoxic and carcinogenic agents [6,15,30,31]. Alkaline comet assay is an excellent tool for detecting genetic damage caused by various compounds [10,32]. Simultaneously, compounds capable of inhibiting genetic damage or inducing repair of damaged DNA have been identified [33–35]. Resveratrol is a powerful antioxidant, which is consumed through the diet and prevents damage to reproductive cells [36]. Ascorbic acid is also supplemented through the diet and its levels depend on the diet of each individual in a range of 5–15 mg/L. It has been reported that ascorbic acid effectively reduces glyphosate-induced toxicity and simultaneously with resveratrol are widely used compounds related to DNA protection and repair [36–40] and could help prevent or inhibit the specific genetic damage caused by glyphosate, reducing the impact that this herbicide has on various organisms in ecosystems at risk of exposure or human populations occupationally exposed to this herbicide.

The objective of this study was to evaluate the genotoxicity of glyphosate and the antigenotoxic effect of ascorbic acid and resveratrol (two powerful antioxidants that could prevent specific genetic damage caused by glyphosate) through pretreatment and simultaneous treatment protocols. The alkaline comet test was used in erythrocytes of *Oreochromis niloticus*, *Ambystoma mexicanum* and human lymphocytes. *O. niloticus* and *A. mexicanum* were used as biomonitors of the genetic impact of glyphosate.

## 2. Materials and Methods

### 2.1. Chemicals and Reagents

Glyphosate (N- (phosphonomethyl) glycine) (CAS 1071-83-6) and the other chemical agents were obtained from Aldrich Chemical Co. (St. Louis, MO, USA). Ethyl methanesulfonate (EMS) (CAS 62-50-0), ascobic acid (CAS 50-81-7) and resveratrol (501-36-0) were obtained from Sigma Chemical Co. (Guadalajara, Jalisco, Mexico), while dimethyl sulfoxide (DMSO, CAS 67-68-5) and disodium salt EDTA (CAS 60-00-4) were obtained from J.T. Baker (Mexico City, Mexico). The concentrations tested for glyphosate were 0.07, 0.7 and 7 (human lymphocytes); 0.03, 0.05 and 0.07 (*Ambystoma mexicanum* and *Oreochromis niloticus* erythrocytes) mM. EMS at 10 mM was used as a positive control and the concentrations of antioxidants used were: ascorbic acid at 5, 10 and 15 mM and resveratrol 0.1, 0.3 and 0.5 mM. These concentrations were previously reported and are capable of granting protection to DNA, therefore, it was decided to test them against the damage induced by glyphosate.

Chemicals and Reagents

Glyphosate (N- (phosphonomethyl) glycine) (CAS 1071-83-6) and the other chemical agents were obtained from Aldrich Chemical Co. (St. Louis, MO, USA). Ethyl methanesulfonate (EMS) (CAS 62-50-0), ascorbic acid (CAS 50-81-7) and resveratrol (501-36-0) were obtained from Sigma Chemical Co. (Guadalajara, Jalisco, Mexico), while dimethyl sulfoxide (DMSO, CAS 67-68-5) and disodium salt EDTA (CAS 60-00-4) were obtained from J.T. Baker (Mexico City, Mexico). The concentrations tested for glyphosate were 0.07, 0.7 and 7 (human lymphocytes); 0.03, 0.05 and 0.07 (*Ambystoma mexicanum* and *Oreochromis*

*niloticus* erythrocytes) mM. EMS at 10 mM was used as a positive control and the concentrations of antioxidants used were: ascorbic acid at 5, 10 and 15 mM and resveratrol 0.1, 0.3 and 0.5 mM. These concentrations were previously reported and are capable of granting protection to DNA, therefore, it was decided to test them against the damage induced by glyphosate.

### 2.2. Obtaining Individuals and Blood Cells

#### 2.2.1. Oreochromis Niloticus

Six fish were obtained from a tilapia aquaculture farm, which supplies organisms for commercial purposes. For 5 days, the fish were heated in 5000 L tanks under natural photoperiod conditions, the constant recycling of running water and with the following physicochemical conditions: zero salinity, temperature $20 \pm 2$ °C, pH $7.0 \pm 0.5$, and dissolved oxygen $8.1 \pm 0.5$ mg/L. In this period, the fish were fed pellets every two days. All animal experiments were complied with the ARRIVE guidelines and were carried out in accordance with the U.K. Animals (Scientific Procedures) Act, 1986 and associated guidelines, EU Directive 2010/63/EU for animal experiments, or the National Institutes of Health guide for the care and use of Laboratory animals (NIH Publications No. 8023, revised 1978). Blood samples were obtained according to Alvarez-Moya et al. [10]: 0.5 mL of blood was collected from the fish gills with the help of a sterile syringe and placed in a tube with 5 mL of phosphate buffer (NaCl 160 mM, $Na_2HPO_4$ 8 mM, $NaH_2PO_4$ 4 mM, EDTA 50 mM, pH 7). They were then centrifuged at 3000 rpm for 10 min and the supernatant was removed. Blood cells were resuspended in 2.5 mL of phosphate buffer and refrigerated at 4 °C until use. The organisms were sacrificed by decapitation [41,42].

#### 2.2.2. Ambystoma Mexicanum

Six axolotls were obtained from an environmental management unit (UMA) located in Zapopan with the proper permits from SEMARNAT. The animals were kept in laboratory conditions according to the recommendations of the "Basic Manual for the captive care of the Xochimilco axolotl (*Ambystoma mexicanum*)" [16]. The axolotls were heated in 100-L tanks and the same conditions were indicated for the fish. All animal experiments were in compliance with the guidance listed above for *Oreochromis niloticus*. Erythrocyte samples were obtained following the steps of the Barriga-Vallejo et al. protocol [43]: a cross-section was made in the branchial apex and 0.1 mL of blood was placed in a 5 mL tube whit phosphate buffer, following the same procedure mentioned above for the fish. No specimens were euthanized, and they were returned to the suppliers.

#### 2.2.3. Human Blood Cells

Prior informed consent and the collaboration of six young students from CUCBA-University of Guadalajara was requested to obtain a blood sample. The inclusion criteria were: not being over 20 years old, not being exposed to environmental pollutants, medications or drugs (information obtained by applying a questionnaire). By annular puncture, 0.1 mL of blood was obtained and it was placed in 5 mL of phosphate solution. The cell pack was obtained and kept refrigerated as mentioned above for fish.

### 2.3. Preparation of Concentrations and Treatment

The glyphosate concentrations to be evaluated were: 0.03, 0.05 and 0.07 in aquatic species and 0.07, 0.7 and 7 mM in human blood cells. As a positive control was used 10 mM EMS. Glyphosate, EMS and phosphate solutions were previously prepared at twice the indicated concentrations, the final concentrations were obtained by mixing 2 mL of the glyphosate solution to be tested with 2 mL of phosphate solution.

#### 2.3.1. Evaluation of Genetic Damage in Blood Cells Exposed to EMS and Glyphosate

First, 50 μL of the samples previously obtained from the individuals to be studied were added to individual tubes containing the solutions with glyphosate and EMS for 2 h at

37 °C. The samples were then centrifuged 3 times at 3000 rpm for 10 min in 4 °C phosphate solution to completely remove debris. Finally, the precipitate was resuspended in 0.5 mL of phosphate solution and preserved at 4 °C to be used with the comet test. Untreated cells in phosphate buffer were used as the negative control.

### 2.3.2. Evaluation of Antigenotoxic Activity

Blood cells were simultaneously exposed to glyphosate, ascorbic acid and resveratrol, following the treatment protocols proposed by Oliveira et al. [44]: pretreatment and simultaneous treatment.

*Pretreatment.* For all the individuals studied, 50 μL of the blood mixture was added in individual tubes at different concentrations of ascorbic acid diluted in phosphate buffer for 2 h. Debris removal was carried out by centrifugation at 3000 rpm for 5 min 3 times. The same was done for resveratrol. For the exposure to glyphosate, the procedure was carried out in the same way as indicated above for the evaluation of genetic damage.

*Simultaneous treatment.* The procedure was carried out in the same way as indicated above for the evaluation of genetic damage, but with the simultaneous addition of phosphate-ascorbic acid buffer with attention to the final concentrations (ascorbic acid 5, 10 and 15 mM and resveratrol 0.1, 0.3 and 0.5 mM). The supernatant was then removed by centrifugation at 3000 rpm for 10 min and the pellet was resuspended in phosphate solution. The procedure was repeated 3 times. Finally, the precipitate was resuspended in 0.5 mL of phosphate solution and kept refrigerated at 4 °C until use.

### 2.4. Comet Test

Slides were covered with 1% normal melting point agarose and subsequently removed to obtain a clean surface. A 0.6% low melting point agarose layer was then placed on the slide. Once solidified, another layer of 0.5% low melting point agarose (90 μL) mixed 10 μL with the previously treated blood cells (genotoxicity and antigenotoxicity) was added, finally, the third layer of 0.5% low melting point agarose. The slides were then placed in lysis buffer (2.5 mM NaCl, 10 mM $Na_2EDTA$, 10 mM Tris-HCl, 1% lauryl sarcosinate, 1% Triton X-100 and 10% DMSO, pH 10) for 24 h at 4 °C. At the end of the lysis, the slides were placed in a horizontal electrophoresis system with electrophoresis buffer (NaOH 300 mM, $Na_2EDTA$ 1 mM) for one hour, then the electrophoresis was carried out for 15 min at 20 V/$cm^2$. The slides were then washed with distilled water and stained with 80 μL of ethidium bromide. Washing was performed by immersion in distilled water for 3 min. Finally, a ten-minute wash was performed to remove excess ethidium bromide.

### Comet Observation and Counting

The observation was carried out on an Axioskop 40 model epifluorescence microscope with a 515–560 nm excitation filter. The tail moment and tail length parameters were measured with the Comet assay system II software. In each of the treatments, 100 comets were counted per individual.

### 2.5. Statistic Analysis

The data obtained were subjected to the statistical program Sigma plot (version 12.0 of "Systat Software, Inc." www.systatsoftware.com (accessed on 17 May 2021), San José, CA, USA) to perform an analysis of variance in Kruskal–Wallis ranges. All the experimental groups were compared with the corresponding negative and positive control by means of the Tukey test with a confidence level of $p < 0.05$.

## 3. Results

### 3.1. Genotoxicity Induced by Glyphosate

Genetic damage of erythrocytes of *Oreochromis niloticus*, *Ambystoma mexicanum* and human lymphocytes exposed to different concentrations of glyphosate (Figure 1). The tail length and tail moment parameters (Figure 1A,B) show genetic damage in erythrocytes of

*Oreochromis niloticus* exposed to glyphosate (0.03, 0.05 and 0.07 mM) and ethyl methane sulphonate (EMS) 10 mM (positive control). All concentrations showed a similar magnitude of genetic and significant differences ($p < 0.05$) with respect to the negative control. Figure 1C,D clearly show genetic damage in erythrocytes of *Ambystoma mexicanum* exposed to glyphosate (0.03, 0.05 and 0.07 mM) and positive control. All concentrations showed significant differences with respect to negative control ($p < 0.05$) and dose–response ratio. Genetic damage was also observed in human lymphocytes (Figure 1E,F), however, the decreased magnitude of damage was the result of extremely long comets that made total tail measurement undetectable; the increased dose caused apparently shorter tails.

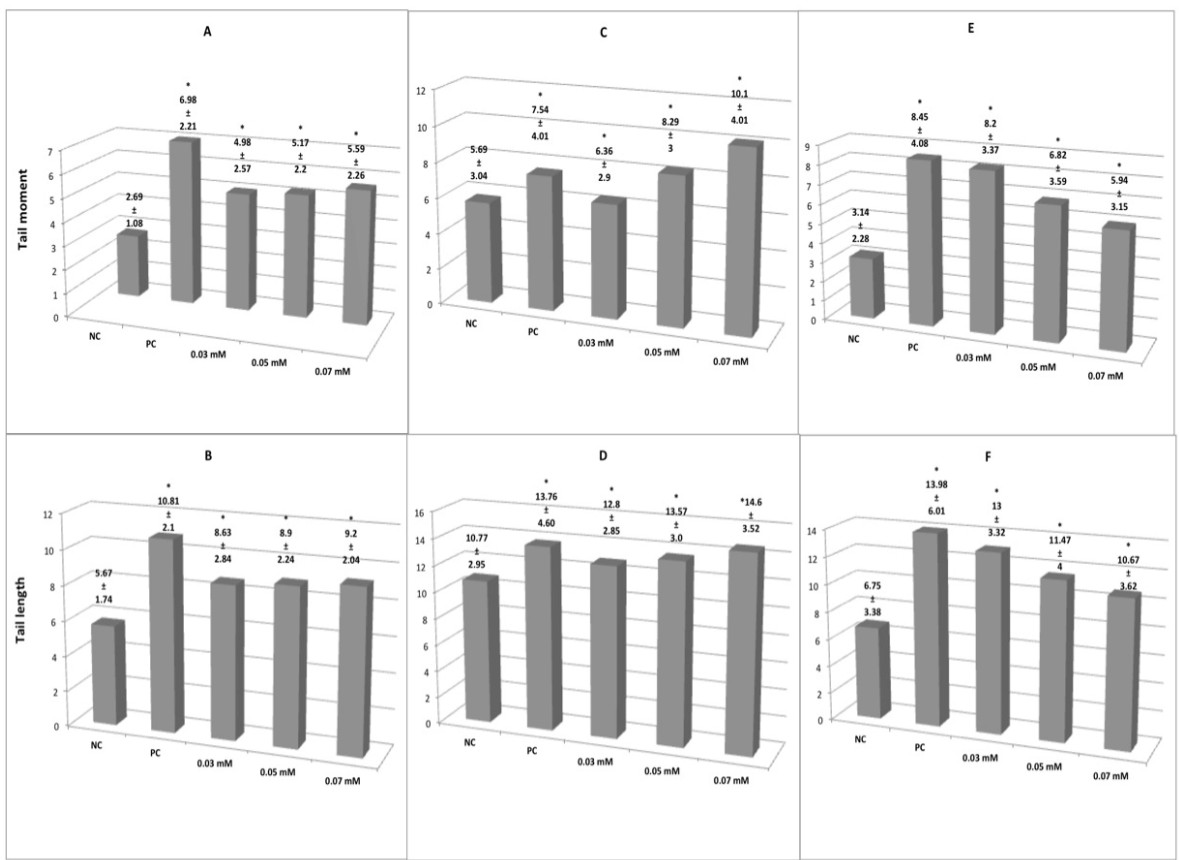

**Figure 1.** Tail moment and tail length in erythrocytes of *Oreochromis niloticus* (**A**,**B**) and *Ambystoma mexicanum* (**C**,**D**) and human lymphocytes (**E**,**F**) exposed to different concentrations of glyphosate. NC, negative control; PC, positive control. All glyphosate concentrations in the three organisms studied were significantly different from NC (*).

*3.2. Antigenotoxic Activity of Ascorbic Acid and Resveratrol in Erythrocytes and Lymphocytes Exposed to Glyphosate*

Antigenotoxic activity of the pretreatment of ascorbic acid and resveratrol in erythrocytes of *Oreochromis niloticus, Ambystoma mexicanum* and human lymphocytes exposed to glyphosate 0.07 mM (Figure 2). In *Oreochromis niloticus,* tail moment as tail length (Figure 2A,B) shows the protective effect of ascorbic acid and resveratrol on DNA; ascorbic acid induces significant reduction of genetic damage ($p < 0.05$) with respect to erythrocytes exposed to glyphosate (0.07 mM), particularly noticeable at concentrations of 5 and 10 mM, 15 mM concentration seems not to be as efficient in reducing genetic damage. Resveratrol 0.1 mM showed a significant protective effect ($p < 0.05$) on the genetic damage induced by glyphosate 0.07 mM in both parameters, however, the increase in the dose does not reduce the genetic damage, it even seems to increase it.

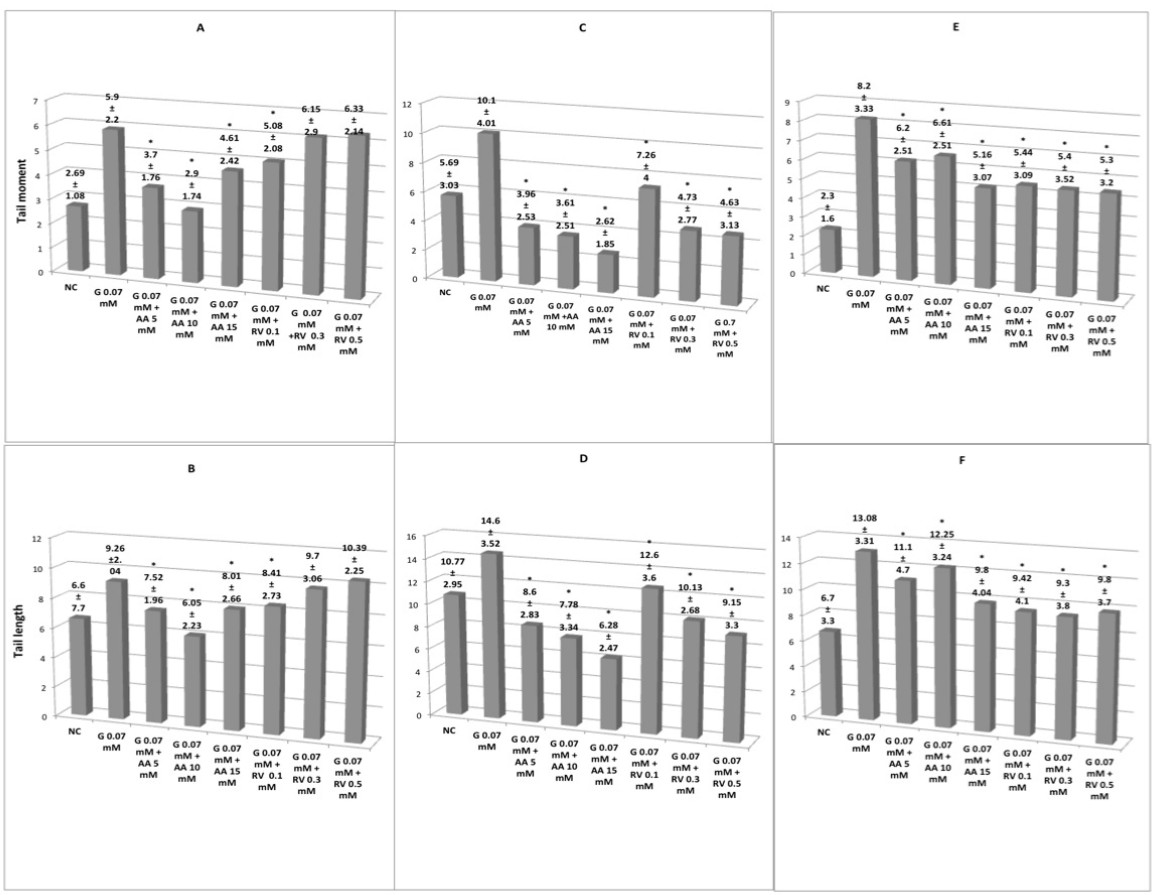

**Figure 2.** Tail moment and tail length in erythrocytes of *Oreochromis niloticus* (**A**,**B**) and *Ambystoma mexicanum* (**C**,**D**) and human lymphocytes (**E**,**F**) exposed to glyphosate 0.07 mM after exposure to ascorbic acid (AA) or resveratrol (RV). NC, negative control; PC, positive control. In (**C**–**F**) G-AA and G-RV concentrations in the three organisms studied were significantly different from G 0.07 mM ($p < 0.05$). In (**A**,**B**), the three concentrations of G-AA were significantly different from G 0.07 mM. In (**A**,**B**), only the concentration G-0.07 mM—RV 0.1 mM was significantly different from G-0.07 mM. * Significant difference ($p < 0.05$) with respect to G 0.07. Glyphosate (G).

In *Ambystoma mexicanum* erythrocytes (Figure 2C,D) and human lymphocytes (Figure 2E,F), ascorbic acid 5, 10 and 15 mM and resveratrol 0.3 and 0.5 mM showed a powerful protective effect in DNA against genetic damage induced by glyphosate 0.07 mM. Using both parameters, genetic damage is reduced to levels similar to that of the negative control, although the protection provided by resveratrol is less than that provided by ascorbic acid. A reduction in damage is observed in direct proportion to the increase in concentration, particularly remarkable for ascorbic acid. In human lymphocytes, all concentrations of ascorbic acid provide similar protection against damage, whereas resveratrol increases protection in proportion to the dose, albeit in a magnitude similar to that of ascorbic acid.

Antigenotoxic activity of the simultaneous treatment of ascorbic acid and resveratrol in erythrocytes of *Oreochromis niloticus*, *Ambystoma mexicanum* and human lymphocytes exposed to glyphosate 0.07 mM (Figure 3). Tail length and tail moment in *Oreochromis niloticus* (Figure 3A,B) show significant difference ($p < 0.05$) of the experimental groups with respect to glyphosate (0.07 mM), the protective effect is evident. Regardless of the dose and the antioxidant, the genetic damage was comparable to that of the negative control or even lower. The magnitude of protection of ascorbic acid is slightly superior to that of resveratrol. Both parameters also show a significant difference ($p < 0.05$) between the experimental groups and glyphosate (0.07 mM) in erythrocytes of *Ambystoma mexicanum* (Figure 3C,D).

No dose–response relationship was observed, although the magnitude of protection of ascorbic acid is slightly superior to that of resveratrol, however, the concentration of resveratrol of 0.3 mM showed no difference with respect to glyphosate in the tail length parameter (Figure 3D). Ascorbic acid and resveratrol showed a powerful protective effect on the DNA of human lymphocytes exposed to glyphosate 0.07 mM (Figure 3E,F). Regardless of the dose, both antioxidants give similar protection to DNA, although the magnitude of protection of resveratrol is slightly higher.

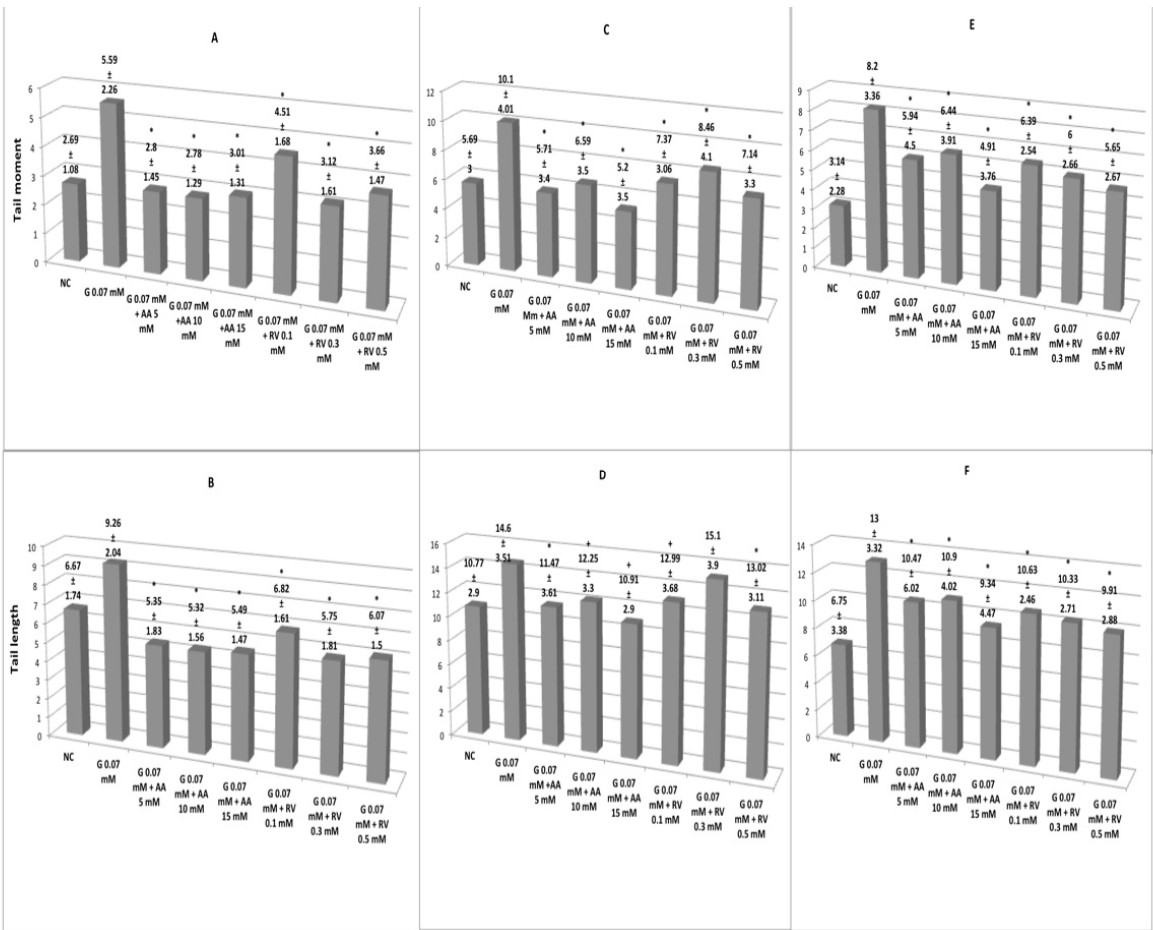

**Figure 3.** Tail moment and tail length in erythrocytes of *Oreochromis niloticus* (**A**,**B**) and *Ambystoma mexicanum* (**C**,**D**) and human lymphocytes (**E**,**F**) exposed to AA or RV simultaneously to 0.07 mM G exposure. Except for the concentration G 0.07 mM + RV 0.03 mM (**D**), all the experimental groups are significantly different ($p < 0.05$) from G 0.07 mM, using both parameters in the three organisms studied. * Significant difference ($p < 0.05$) with respect to G 0.07.

## 4. Discussion

Glyphosate genotoxicity is highly controversial: its genotoxicity has been reported [45,46]. In contrast, Farthing et al. [47] mention that the application of glyphosate has no significant impact on the richness and diversity of species. Other reports indicate that chemically pure glyphosate lacks genotoxicity or that the genotoxicity of glyphosate is due to the presence of adjuvants employed in commercial glyphosate-based formulations [48–50] which are considered inert, however, some of them are more toxic or genotoxic than the pesticide [45]. Due to the increasing use of glyphosate, there is a need to use bioassays to evaluate their genotoxic effects [31].

In the present work, the impact of glyphosate on wild organisms such as *Oreochromis niloticus* and *Ambystoma mexicanum* is clear, at least in concentrations similar to those used by other authors [5,21], albeit lower than the sublethal concentrations used by Poletta et al. [15]

(50, 100, 200, 300,400, 500, 750, 1000, 1250 and 1750 g/egg). This impact has been reported for other organisms [6]. *Ambystoma mexicanum* is an endemic species and this work constitutes the first report on the genotoxic effect of glyphosate in this species by using the comet test. Robles-Mendoza et al. [51] noted the presence of agrochemicals in the natural habitat of *Ambystoma mexicanum*, as indicated by our data at the concentrations studied and is particularly visible at the 0.07 mM concentration, therefore the dangerousness of glyphosate could affect the genomic stability of these populations. Alvarez Moya et al. [10] calculated that the environmental concentrations of glyphosate in a human population, adjacent to fields under FAENA fumigation conditions, ranged from 0.7 to 0.0007 mM, therefore, we used approximate glyphosate concentrations.

The genotoxic activity of glyphosate was also evident in human lymphocytes and agrees with what was reported by other authors who used variable concentrations [2,10,21]. The reduction in genetic damage observed in human lymphocytes after using high concentrations of glyphosate is only apparent: the excessive induced genetic damage causes excessive and undetectable migration in the tail length as was previously argued by other authors [52] which explains the apparent decrease in genetic damage despite the increase in the concentration of glyphosate, at least in humans. In *Oreochromis niloticus* and *Ambystoma mexicanum* glyphosate, 0.07 mM of glyphosate induced considerable genetic damage and is adequately quantified by the software. Human blood cells appear to be more sensitive. Our data showed the genotoxic effect of glyphosate, but there were variations depending on the species and the concentration studied; these variations have been observed in other organisms, which could explain some contradictions regarding the genotoxicity of glyphosate [19].

The consumption of antioxidants can contribute to the maintenance of the genetic integrity of organisms exposed to genotoxins [37,53]. Ascorbic acid and resveratrol are two powerful antigenotoxic agents that have the ability to prevent genotoxin-induced genetic damage [54–56]. Organisms acquire these antioxidants through diet, so diet type may be associated with protection against glyphosate-induced genetic damage. Knowing whether these antioxidants have antigenotoxic capacity against glyphosate is relevant regardless of the presence of these compounds in free form in the environment. It is considered a first step in possible future applications. Regardless of the exposure protocol used and species studied, our data strongly indicate that these antioxidants could help preserve DNA integrity in organisms exposed to glyphosate. The protection by ascorbic acid and resveratrol against genetic damage caused by glyphosate is particularly notable in *A. mexicanus*, similar results were reported by de Jesús et al. [5], however, in this species, the decrease in genetic damage is proportional to the concentration of antioxidants. Resveratrol also showed concentration-proportional protection, as previously reported [36]. The protective effect of ascorbic acid is evident in *O. niloticus* and *H. sapiens*, as reported in human cells cryopreserved at higher concentrations [36].

There is difficulty in definitively establishing the genotoxicity of glyphosate; less is known about the molecular mechanism by which glyphosate causes DNA damage [57], however, Woźniak et al. [46] reported that glyphosate can induce DNA single and double strand-breaks and purines and pyrimidines oxidation, but not adducts, and suggest that the mechanism of DNA damage induced by Roundup occurred through ROS-mediated effects. Therefore, antioxidants such as ascorbic acid and resveratrol supplied through the diet will play an important role in preventing genetic damage caused by glyphosate, mainly in occupationally exposed human population and in sensitive wild species. Our data show that ascorbic acid was able to prevent (pretreatment) and inhibit (simultaneous treatment) the specific genetic damage caused by glyphosate in erythrocytes and lymphocytes of the three studied species, and they confirm the antigenotoxic activity of ascorbic acid [58].

Although the mode of action of ascorbic acid and resveratrol is not very clear [40], it is possible that pretreatment with ascorbic acid or resveratrol increases the efficiency of antioxidant protection systems and reduces the genotoxic potential of glyphosate. Pretreatment also strengthens cell protection and repair systems [59]. Our data indicate that

DNA protection is effective, since the genetic damage observed after pretreatment with ascorbic acid is similar to that observed in negative controls, which suggests a chemical blockade of glyphosate and stimulation of repair systems, as has been reported for other compounds [60]. The same was observed for resveratrol in humans. As we were unable to define which protocol provides greater protection, it is not ruled out that both ascorbic acid and resveratrol increase the efficiency of both systems, antioxidant protection and DNA. Wozniak et al. [46] suggests that the genetic damage induced by glyphosate is associated with oxidative stress and an increase in reactive species (ROS). Ascorbic acid and resveratrol reduce genetic damage caused by oxidative stress [55], therefore, the reduction in genetic damage caused by glyphosate in the blood cells studied here is caused by the chemical neutralization of ROS by the action of ascorbic acid and resveratrol, at least in simultaneous treatment. The data with the pre-treatment also indicate a strong protective effect of ascorbic acid and resveratrol, probably associated with the strengthening of antioxidant protection systems, but we lack data to confirm this. The presence of antioxidant molecules prevalent in the nucleoplasm with the ability to chemically neutralize ROS cannot be ruled out either.

Resveratrol also prevents and inhibits glyphosate-induced genetic damage, however, pretreatment in the erythrocytes *Oreochromis niloticus* and simultaneous treatment in the erythrocytes of *Ambystoma mexicanum* showed contrasting results: Pretreatment in the erythrocytes *Oreochromis niloticus* suggests that the higher concentration of resveratrol causes genetic damage. Probably, the concentrations used are high for the organism and are even capable of inducing genetic damage; the above agrees with the reported works by various authors, who point out that resveratrol can induce genetic damage in various conditions [61,62], however, the lowest concentration used in this study was, effectively, able to reduce the genetic damage generated by glyphosate, as reported by Jia et al. [63]. Our data suggest that concentrations between 0.1 and 0.3 mM of resveratrol are adequate to protect the integrity of DNA in at least this organism; however, in erythrocytes from *Ambystoma mexicanum* and human lymphocytes, resveratrol in all concentrations used exerts a powerful protective effect in glyphosate-induced genetic damage. Pretreatment seems to strengthen cell protection and repair systems [59,60].

In the case of *Ambystoma mexicanum*, resveratrol has a notable protective difference with respect to ascorbic acid; there are probably metabolic differences between organisms that affect the protective effect of resveratrol [64]. Our results on resveratrol agree with various reports which indicate the protective effect of resveratrol in different organisms [65–67]. The antigenotoxic capacity of resveratrol is consistent with data from Chen et al. [68] who reported inhibition by the effect of sodium arsenite, a pesticide related to the appearance of tumors. Santo et al. [33] mentioned that glyphosate-induced genetic damage could be inhibited by the use of plant extracts in zebrafish (Danio rerio); these data are consistent with our results for ascorbic acid and resveratrol. In light of these results, although the consumption of antioxidants is a useful tool against the genotoxicity of glyphosate, it is clear that it would be better to opt for pesticides that are less dangerous to DNA.

## 5. Conclusions

Glyphosate showed high genotoxic capacity in the blood cells of the organisms studied, however, the presence of ascorbic acid and resveratrol significantly reduced its genotoxic potential in both protocols: pretreatment and simultaneous treatment, so these compounds can increase the efficiency of the antioxidant protection and of the DNA repair system. Ascorbic acid and resveratrol in human lymphocytes and blood cells of *Oreochromis niloticus* and *Ambystoma mexicanum* reduced glyphosate-induced genetic damage until basal level. Regarding the sensitivity of glyphosate, variation was observed among the species. The frequent consumption of these two antioxidants can play an important role in maintaining the genetic stability of the species studied. The genotoxicity of glyphosate in *Ambystoma mexicanum* erythrocytes with the alkaline comet test is reported for the first time.

**Author Contributions:** Conceptualization, C.A.-M. and M.R.-S.; methodology, A.G.S.-L., R.R.-V.; validation, A.G.S.-L., C.A.-M. and M.R.-S.; formal analysis, A.G.S.-L.; investigation, C.A.-M.; resources, C.A.-M. and A.R.V.-A.; writing—original draft preparation, M.R.-S. and A.R.V.-A.; writing—review and editing, C.A.-M., M.A.R.-L. and M.R.-S.; visualization, C.A.-M.; supervision, C.A.-M.; project administration, M.A.R.-L.; funding acquisition, A.R.V.-A. All authors have read and agreed to the published version of the manuscript.

**Funding:** This research was funded by UNIVERSIDAD DE GUADALAJARA, México, grant number: UNIVERSIDAD DE GUADALAJARA-BCyM/258/2021.

**Institutional Review Board Statement:** The study was conducted according to the guidelines of the Declaration of Helsinki, and approved by the Institutional Review Board of DBCyM (DB-CyM/200/2020; 11 February 2020).

**Informed Consent Statement:** Informed consent was obtained from all subjects involved in the study.

**Data Availability Statement:** The data presented in this study are available on request from the corresponding author.

**Acknowledgments:** Authors are grateful for the collaboration of CONACYT, who provided financial support to the students involved in this study.

**Conflicts of Interest:** The authors declare no conflict of interest.

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
