# Peer review of "Antigenotoxic Effect of Ascorbic Acid and Resveratrol in Erythrocytes of Ambystoma mexicanum, Oreochromis niloticus and Human Lymphocytes Exposed to Glyphosate"

_cimb, doi:10.3390/cimb44050151_

Round 1

Reviewer 1 Report

Line 26; „...The genotoxic activity of glyphosate was evident and significant...“ Pleae, explain which concentrations provoke this significant difference.

Line 28-30, „The reduction in genetic damage observed in human lymphocytes exposed to high concentrations of glyphosate is not real: excessive genetic damage was associated with undetectable excessive tail 31 migration length.“. Please explain what do you consider by high concentrations and why this result is not real. What does number 31 stands for?

Introduction section

Please explain what mechanism is responsible for glyphosate genotoxicity?

Please explain do reservatrol and ascorbic acid reach human lymphocytes in their parental form? At which concentration these compounds can be found in the blood? Is gyphosate metabolised prior it reaches the blood? At which concentrations this compound (in its parental form) can be found in the blood?

Material and methods section:

Autors should explain if they have tested citotoxicity of glyphosate, ascorbic acid and reservatrol and their mixture. If not, how they can be sure that tested concentrations do not have toxic effect on the cells analysed in comet assay?

Line 137, „ The glyphosate concentrations to be evaluated were: 0.03, 0.05, and 0.07 in aquaticspecies and 0.07, 0.7, and 7 mM in human blood cells.“ What was the rationale to test these concentrations on aquatic species and human blood cells respectivelly?

Please indicate what concentrations of ascorbic acid and reservatrol were tested and what was the rationale to test these concentrations?

Results:

Figure 1. Please explain decrease of tail intensity in lymphocytes at the highest concentration of glyphosate tested?

Please delete the titles in the figures. Figures are described in the text below the figures.

Discussion section:

Line 287, „...Due to the increasing use of pesticides..“ glyphosate is herbicide and not pesticide so Authors should take it into consideration.

Please discuss and comment what are the expected concentrations of glyphosate in the environment. Speaking of exotoxicity and protective effect of reservatrol and ascorbic acid against glyphosate in aquatic organisms, what is the posibility that these two compounds are found in the environment? Are these compounds metabolized in aquatic organisms?

Please comment in which form it is expected to found glyphosate, ascorbic acid and reservatrol in human blood? At which concentrations? According to that, please comment used concentrations and results obtained in this paper.

Reviewer 2 Report

The paper addresses the antigenotoxic effect of ascorbic acid and resveratrol in cell models of genotoxicity induced by glyphosate. In this tudy, erythrocytes of Oreochromis  niloticus, Ambystoma mexicanum, and human lymphocytes as used.

The authors made some modifications to the manuscript according to the reviewers' comments.

However, the following aspects should be taken into account in order to improve the quality of the paper:

  • The objectives of the article (end of the introduction) must be clear and well structured. Please improve this part.
  • The figures must be improved in their presentation, content and quality (blurred).
  • Statistical analysis should be included in the figures
  • Figures must be distributed in the manuscript following the citation in the text (not in a separate section).
  • Please review all temperature units (°C)
  • The results of the paper must be related to the results of other authors. This is not done in the revision process. ("...was reported by other authors" is not to discuss the results with those in the literature).
  • The conclusions must be improved.

Round 2

Reviewer 1 Report

I have reviewed revised version of the manuscript. The Authors have made significant corrections of the text and explained some critical points. In this form I think this paper could be accepted for publication.

This manuscript is a resubmission of an earlier submission. The following is a list of the peer review reports and author responses from that submission.

Round 1

Reviewer 1 Report

I have reviewed an article: "Antigenotoxic effect of ascorbic acid and resveratrol in erythrocytes of Ambystoma mexicanum, Oreochromis niloticus and human lymphocytes exposed to glyphosate.

General comment: please do not use abbreviations such as G, AA, RV etc.

Specific comment:

Introduction section: Please indicate what is the half-life of gyphosate under environmental conditions.

What is the mechanism responsible for glyphosate genotoxicity?

MM section: Please explain why did you test glyphosate in concentration range 0.03-0.07 and 0.07-0.7 respectively? Do these concentrations have any connection to the concentrations that can be found as residues and contaminants in the environment?

Also, please explain what is the significance of tested concentrations of ascorbic acid and resveratrol?

Discussion section: Please comment why adjuvants that are added to commercial herbicides possess genotoxic activity.

Please discuss different countries' politics considering banning the use of glyphosate. Could you comment it into a light of your experimental data?

Author Response

Comments and Suggestions for Authors

1          General comment: please do not use abbreviations such as G, AA, RV etc.

Answer: Abbreviations like G, AA RV and others have been removed from the text.

2          Introduction section: Add information on the half-life of gyphosate under environmental conditions.

Answer: Done

3          What is the mechanism responsible for glyphosate genotoxicity?

Answer: Done

Exist a vigorous debate about genotoxicity and carcinogenicity of glyphosate by scientific and regulatory agencies, so, information on the mechanism of glyphosate genotoxicity is scarce or non-existent. An excerpt from the article by Nagy et al. (2019)   is added:

Information about the mechanism responsible for glyphosate genotoxicity has been added in the introduction section.In recent years, genotoxicity and carcinogenicity of glyphosate has been under vigorous – and so far inconclusive – review and debate by scientific and regulatory agencies. The International Agency for Research on Cancer (IARC) classified glyphosate as “probably carcinogenic to humans” (Group 2A) in 2015 based on limited epidemiological evidence in humans (for non-Hodgkin lymphoma), sufficient evidence in experimental animals with strong evidence for two carcinogenic mechanism, genotoxicity and oxidative stress (IARC, 2015; Portier et al., 2016). In the same year, the European Food Safety Authority (EFSA) concluded that glyphosate is unlikely to be genotoxic (i.e. damaging to DNA) or to pose a carcinogenic threat to humans because its experts did not find evidence from neither epidemiological nor animal studies on the causality between exposure to glyphosate and the development of cancer in humans (EFSA, 2015a). However, in agreement with the IARC evaluation, EFSA has confirmed that glyphosate induces oxidative stress, nevertheless emphasized that the ability of inducing oxidative stress alone is not sufficient to classify glyphosate as carcinogenic (Portier et al., 2016). EFSA also criticized the assessment from IARC for not including some studies in the evaluation process which was one of the reasons for reaching different conclusions. In 2016, the U.S. Environmental Protection Agency (US EPA) also 

declared that glyphosate is not likely to be carcinogenic to humansas a result of a comprehensive review of dietary, residential/ non-occupational, aggregate, and occupational human exposures, as well as an in-depth review of the glyphosate cancer database, including data from epidemiological, animal carcinogenicity, and genotoxicity studies (US EPA, 2016). The controversy over the classification of glyphosate's genotoxicity and carcinogenicity is likely to be explained by the substantially different evaluation methodologies of the authorities and makes it difficult to bring us closer to a final conclusion on the DNA. damaging or cancer-causing ability of this herbicide.

Information about this debate has been added in the introduction section.

4          MM section: Please explain why did you test glyphosate in concentration range 0.03-0.07 and 0.07-0.7 respectively? Do these concentrations have any connection to the concentrations that can be found as residues and contaminants in the environment?

Answer: Indeed, it is related to environmental concentrations. DNA damage was previously reported in people occupationally exposed to glyphosate and compared with in vivo studies (Alvarez. Moya et al., 2011, 2014). Genetics and Molecular Biology, 37, 1, 105-110 (2014); Genetics and Molecular Biology, 34, 1, 127-130 (2011).

5          Also, please explain what is the significance of tested concentrations of ascorbic acid and resveratrol?

Answer: Done

These concentrations were previously reported and are capable of granting protection to DNA, therefore it was decided to test them against the damage induced by glyphosate. In the case of resveratrol, we do not know if these are supplied in a normal human diet, however, in the case of ascorbic acid, these concentrations may be even higher, depending on the type of diet.

6          Discussion section: Please comment why adjuvants that are added to commercial herbicides possess genotoxic activity

Adjuvants are added to pesticides to improve their performance. They are considered inert, however some of them are more toxic or genotoxic than the pesticide. It is recognized that POEAs or GBHs containing POEAs are more toxic in acute and chronic toxicity tests than glyphosate alone,

7          Please discuss different countries' politics considering banning the use of glyphosate. Could you comment it into a light of your experimental data?

Answer: Done in Discussion section

In Europe (France, Germany, Portugal, Spain, Italy) there are citizen organizations that fight for the prohibition of glyphosate. In some Latin American countries, the damage caused by glyphosate has been devastating and documented. Mexican President Andrés Manuel López Obrador has given farmers until 2024 to stop using glyphosate. On December 31, the country published a "final decree" however Monsanto's owner Bayer AG and industry lobbyist CropLife America have been working closely with US officials to pressure Mexico to leave. their intention to ban glyphosate.

In light of these results, although the consumption of antioxidants is a useful tool against the genotoxicity of glyphosate, it is clear that it would be better to opt for pesticides that are less dangerous to DNA.

Reviewer 2 Report

The study by Álvarez-Moya et al on "Antigenotoxic effect of ascorbic acid and resveratrol in erythrocytes of Ambystoma mexicanum, Oreochromis niloticus and human lymphocytes exposed to glyphosate" present a brief overview on the genotoxic effects of  ascorbic acid and resveratrol. Although, the manuscript is trying to address an important issue of the  Glyphosate, a controversial herbicide, however, the study fails to prove the novelty and relevance of the research. 

In addition, the experiments presented are week and the data representation is also not up to the mark and need to be done is a more suitable manner for better understanding of the readers. Studies are already available describing that the ascorbic acid (doi.org/10.1191/0960327104ht508oa) and resveratrol both have antigenotoxic effects, hence it is not clear that was was the basis for selection of these two compounds (doi: 10.1016/j.fct.2013.05.030). 

Therefore this reviewer is not in favour for the acceptance of the manuscript untill more comprehensive experimental evidences and novelty aspect will be included. 

Author Response

REWIEWER 2

Comments and Suggestions for Authors

The study by Álvarez-Moya et al on "Antigenotoxic effect of ascorbic acid and resveratrol in erythrocytes of Ambystoma mexicanum, Oreochromis niloticus and human lymphocytes exposed to glyphosate" present a brief overview on the genotoxic effects of  ascorbic acid and resveratrol. Although, the manuscript is trying to address an important issue of the  Glyphosate, a controversial herbicide, however, the study fails to prove the novelty and relevance of the research. 

In addition, the experiments presented are week and the data representation is also not up to the mark and need to be done is a more suitable manner for better understanding of the readers. Studies are already available describing that the ascorbic acid (doi.org/10.1191/0960327104ht508oa) and resveratrol both have antigenotoxic effects, hence it is not clear that was was the basis for selection of these two compounds (doi:10.1016/j.fct.2013.05.030). 

Therefore this reviewer is not in favour for the acceptance of the manuscript untill more comprehensive experimental evidences and novelty aspect will be included. 

Answer: Dear reviewer, your opinion is respectable.

There are three points that may not have been adequately appreciated:

1 This study is not intended to discover the genotoxicity of glyphosate; it is known. The genetic damage induced by glyphosate in two wild species was investigated and related to a possible impact on the environment and on human health resulting from the use of this pesticide.

2 The protective capacity of resveratrol and ascorbic acid (two antioxidants widely consumed through the diet and with known antigenotoxic capacity) is known. The objective of this work is to investigate whether these antioxidants have a protective capacity against specific genetic damage induced by glyphosate.

3 Ambystoma mexicanum, is an endemic species of the state of Mexico, Mexico. The extensive use of glyphosate has led to the contamination of the waters (Xochimilco) where this species lives, of which no studies have been carried out to evaluate its genetic integrity.

Submission Date

13 December 2021

Date of this review

13 Jan 2022 05:07:49

Reviewer 3 Report

Title: Antigenotoxic effect of ascorbic acid and resveratrol in erythrocytes of Ambystoma mexicanum, Oreochromis niloticus and human lymphocytes exposed to glyphosate

The paper addresses the antigenotoxic effect of ascorbic acid and resveratrol in cell models of genotoxicity induced by glyphosate. In this tudy, erythrocytes of Oreochromis  niloticus, Ambystoma mexicanum, and human lymphocytes as used.

The work in general is interesting, but the results need to be better described and discussed.

The following aspects should be taken into account in order to improve the quality of the paper:

  1. Please use the CIMB template not the IJMS.
  2. Improve the structure and format of the document.
  3. Extensive editing of English language and style required.
  4. Introduction need to reformulated and improved.
  5. The abbreviation for glyphosate (G) should be used in the body of the manuscript (eg, line 34, 37, 39, 41…..).
  6. Abstract: Important information referred to in the body of the document must be included in the abstract. The abstract must be improved.

Results:

  1. Line 65: whats means PC? Positive control?
  2. The figures must be improved.
  3. Some differences between groups are not visibly different. I recommend using another type of graphical representation to improve the visibility of the results.
  4. Results did not demonstrate a dose-response effect on the genotoxic activity of glyphosate. How do you explain these results?
  5. Why the authors selected the G dose of 0.07M for further tests? The lower concentrations of the pesticide present significant differences compared to NC.
  6. * Different letters indicate significant differences… (line 100). Must be included in the figure caption
  7. In general, the results need a better description in the manuscript.

Discussion:

  1. Line 119-121: Previously reported in the introduction. Please do not repeat information.
  2. The results of the research paper must be related to the results of other authors. This is not done in this work. Please consider this for further submission.

Materials and Methods:

The methodology should be improved and specified for each protocol. Some important parameters are not specified or referenced.

  1. Section 4.1 must be improved, and other chemicals should be included.
  2. Line 190: review the phosphate buffer composition (NaCl 160 mM, Na2HPO4 8 mM, Na2HPO4 4 mM, EDTA 50 mM, pH 7).
  3. Replace `ml´ by `mL´ throughout the manuscript.
  4. Line 193-194: Please explain this sentence. In my opinion, it's not essential information.
  5. Line 197: What does SEMARNAT mean?
  6. Section 4.2.3: Human blood cells. Were these experiments, and the previous ones, subject to a pre-assessment by an ethics committee? This must be referenced in the document (reference from the ethical committee of the research center or the university).
  7. Section 4.3: What are the parameters used to select glyphosate and EMS concentrations for each model?
  8. Section 4.3.1/ 4.3.2. Please justify the use of this protocol. Do you have any reference? Why use temperatures of 4ºC?
  9. Section 4.4.: 4 °C; 20 V/cm2; section 2.5 (should be 4.4.1., correct?)
  10. Section 2.6. (page 8), should be 4.5.  

  1. The conclusions must be improved.

Author Response

REWIEWER 3       THANKS FOR YOUR VALUABLE HELP

Title: Antigenotoxic effect of ascorbic acid and resveratrol in erythrocytes of Ambystoma mexicanum, Oreochromis niloticus and human lymphocytes exposed to glyphosate

The paper addresses the antigenotoxic effect of ascorbic acid and resveratrol in cell models of genotoxicity induced by glyphosate. In this tudy, erythrocytes of Oreochromis  niloticus, Ambystoma mexicanum, and human lymphocytes as used.

The work in general is interesting, but the results need to be better described and discussed.

The following aspects should be taken into account in order to improve the quality of the paper:

  1. Please use the CIMB template not the IJMS. DONE
  2. Improve the structure and format of the document. DONE
  3. Extensive editing of English language and style required. EDITORIAL HELP REQUESTED
  4. Introduction need to reformulated and improved. DONE
  5. The abbreviation for glyphosate (G) should be used in the body of the manuscript (eg, line 34, 37, 39, 41…..).

Answer: On the recommendation of another reviewer, abbreviations were omitted.

  1. Abstract: Important information referred to in the body of the document must be included in the abstract. The abstract must be improved. DONE

Results:

  1. Line 65: whats means PC? Positive control? DONE
  2. The figures must be improved: DONE
  3. Some differences between groups are not visibly different. I recommend using another type of graphical representation to improve the visibility of the results.
  4. Answer: The figures were improved.

  1. Results did not demonstrate a dose-response effect on the genotoxic activity of glyphosate. How do you explain these results?

Answer: The excessive induced genetic damage causes excessive and undetectable migration in the flow of comets as was previously argued by other genotoxic [52]. This was mentioned in the discussion section.

  1. Why the authors selected the G dose of 0.07M for further tests? The lower concentrations of the pesticide present significant differences compared to NC.

Answer: It is related to environmental concentrations. DNA damage was previously reported in people occupationally exposed to glyphosate and compared with in vivo studies (Alvarez. Moya et al., 2011, 2014). Genetics and Molecular Biology, 37, 1, 105-110 (2014); Genetics and Molecular Biology, 34, 1, 127-130 (2011).

  1. * Different letters indicate significant differences… (line 100). Must be included in the figure caption: DONE
  2. In general, the results need a better description in the manuscript.: DONE

Discussion:

  1. Line 119-121: Previously reported in the introduction. Please do not repeat information. DONE
  2. The results of the research paper must be related to the results of other authors. This is not done in this work. Please consider this for further submission.

Answer: Thank you very much, this point is very important and we worked on improving this section.

Materials and Methods:

The methodology should be improved and specified for each protocol. Some important parameters are not specified or referenced.

  1. Section 4.1 must be improved, and other chemicals should be included. DONE
  2. Line 190: review the phosphate buffer composition (NaCl 160 mM, Na2HPO4 8 mM, Na2HPO4 4 mM, EDTA 50 mM, pH 7): DONE

  1. Replace `ml´ by `mL´ throughout the manuscript: DONE
  2. Line 193-194: Please explain this sentence. In my opinion, it's not essential information. DONE
  3. Line 197: What does SEMARNAT mean?: Minsitry of environment (Secretaria del medio ambiente y recursos naturales)
  4. Section 4.2.3: Human blood cells. Were these experiments, and the previous ones, subject to a pre-assessment by an ethics committee? This must be referenced in the document (reference from the ethical committee of the research center or the university).

Answer: Yes this informatión is showed in section: Institutional Review Board Statement: The study was conducted according to the guidelines of the Declaration of Helsinki, and approved by the Institutional Review Board of DBCyM (DBCyM/200/2020; 11/02/2020).

  1. Section 4.3: What are the parameters used to select glyphosate and EMS concentrations for each model?

Answer: the concentrations of glyphosate reported in the literature as genotoxic are extremely variable. We use concentrations previously reported and related to the species to be studied. In the case of EMS, we use the concentration that we have observed to be effective in inducing genetic damage and that has also been reported.

  1. Section 4.3.1/ 4.3.2. Please justify the use of this protocol. Do you have any reference? Why use temperatures of 4ºC?

Answer: Thanks for the comment, it was a mistake. We used the procedure of Nagy et al.

Nagy, Károly, et al. "Comparative cyto-and genotoxicity assessment of glyphosate and glyphosate-based herbicides in human peripheral white blood cells." Environmental research179 (2019): 108851.

Nagy, Károly, et al. "Micronucleus formation induced by glyphosate and glyphosate-based herbicides in human peripheral white blood cells." Frontiers in public health 9 (2021).

  1. Section 4.4.: 4 °C; 20 V/cm2; section 2.5 (should be 4.4.1., correct?) :DONE
  2. Section 2.6. (page 8), should be 4.5. : DONE

  1. The conclusions must be improved. DONE

Round 2

Reviewer 2 Report

Although the authors have done changes in their manuscript. However, still his reviewer is not fully satisfied with the presentation of data and novelty of the work. The reviewer leave the acceptance of the article on the enlightenment of editor. 

Author Response

Dear reviewer 2, thank you for your valuable comments. We work hard on the manuscript during the days given by the editor. The points that you wisely indicated were strengthened. On this occasion the writing of the English language was improved. Aspects of the experimental design and methodology were also highlighted, which were not clearly described as you intelligently mentioned. Simultaneously the introduction was increased and improved.

Again thank you very much.

Reviewer 3 Report

The authors generally made the modifications proposed by the reviewers, adding scientific evidence to the work.

Some minor comments: 

The sentence should be revised:

- "which could affect the genomic stability of these populations" (line 28)

- “The reduction in genetic damage observed in human lymphocytes after using high concentrations of glyphosate is only apparent: the excessive induced genetic damage causes excessive and undetectable migration in the tail length.” (Line 28-.31).

- Line 77-80: improve this sentence.

- Line 244: "In humans, all concentrations ...". Improve this sentence. In this work, no studies were carried out with humans, only human cells (lien 363 also).

- Line 258-263: improve this sentence.

- Figures must follow their citation in the text and not in a separate section.

- Line 357: improve this sentence.

- Line 391-393: improve this sentence.

- Conclusions can be further refined. 

Author Response

Reviewer 3

Open Review

Thank you for your valuable comments. We work hard on the manuscript during the days given by the editor. The points indicated have been corrected.

Again thank you very much.

Some minor comments: 

The sentence should be revised:

- "which could affect the genomic stability of these populations" (line 28): DONE

- “The reduction in genetic damage observed in human lymphocytes after using high concentrations of glyphosate is only apparent: the excessive induced genetic damage causes excessive and undetectable migration in the tail length.” (Line 28-.31). DONE

- Line 77-80: improve this sentence. DONE

- Line 244: "In humans, all concentrations ...". Improve this sentence. In this work, no studies were carried out with humans, only human cells (lien 363 also). DONE

- Line 258-263: improve this sentence. DONE

- Figures must follow their citation in the text and not in a separate section. DONE

- Line 357: improve this sentence. DONE

- Line 391-393: improve this sentence. DONE

- Conclusions can be further refined. DONE

Submission Date

13 December 2021

Date of this review

08 Feb 2022 16:12:16